# Circulating Chemerin Levels, but not the *RARRES2* Polymorphisms, Predict the Long-Term Outcome of Angiographically Confirmed Coronary Artery Disease

**DOI:** 10.3390/ijms20051174

**Published:** 2019-03-07

**Authors:** Leay Kiaw Er, Lung-An Hsu, Jyh-Ming Jimmy Juang, Fu-Tien Chiang, Ming-Sheng Teng, I-Shiang Tzeng, Semon Wu, Jeng-Feng Lin, Yu-Lin Ko

**Affiliations:** 1The Division of Endocrinology and Metabolism, Department of Internal Medicine, Taipei Tzu Chi Hospital, Buddhist Tzu Chi Medical Foundation, New Taipei City 23142, Taiwan; leay29010@gmail.com; 2School of Medicine, Tzu Chi University, Hualien 97004, Taiwan; jf6520@yahoo.com.tw; 3The First Cardiovascular Division, Department of Internal Medicine, Chang Gung Memorial Hospital and Chang Gung University College of Medicine, Taoyuan 33305, Taiwan; hsula@adm.cgmh.org.tw; 4Cardiovascular Center and Division of Cardiology, Department of Internal Medicine, National Taiwan University Hospital city, Taipei 10002, Taiwan; P91421019@ntu.edu.tw (J.-M.J.J.); futienc@ntuh.gov.tw (F.-T.C.); 5Taipei, Taiwan and National Taiwan University College of Medicine, Taipei 10002, Taiwan; 6Cardiovascular Center and Division of Cardiology, Fu-Jen Catholic University Hospital, New Taipei city 24352, Taiwan; 7Department of Research, Taipei Tzu Chi Hospital, Buddhist Tzu Chi Medical Foundation, New Taipei city 23142, Taiwan; vincent@tzuchi.com.tw (M.-S.T.); xdd05082@tzuchi.com.tw (I.-S.T.); 8Department of Life Science, Chinese Culture University, Taipei 11114, Taiwan; semonwu@yahoo.com.tw; 9Cardiovascular Center and Division of Cardiology, Department of Internal Medicine, Taipei Tzu Chi Hospital, Buddhist Tzu Chi Medical Foundation, New Taipei city 23142, Taiwan

**Keywords:** genome-wide association study, chemerin, *RARRES2* gene, coronary artery disease, all-cause mortality

## Abstract

Chemerin, a novel adipokine, has been associated with metabolic, inflammatory, and atherosclerotic diseases. We aimed to determine the genetic basis of chemerin levels by conducting a genome-wide association study (GWAS) and to investigate the role of *RARRES2* polymorphisms and circulating chemerin levels in the long-term outcome of coronary artery disease (CAD). A total of 2197 participants from the Taiwan Biobank (TWB) were recruited for the GWAS analysis, and 481 patients with angiographically confirmed CAD were enrolled for long-term outcome analysis. One locus of genome-wide significance with a single independent association signal was identified in the GWAS for chemerin levels with the peak association at the *RARRES2* gene promoter region polymorphism rs3735167 (*p* = 2.35 × 10^−21^). In the CAD population, borderline significance was noted between *RARRES2* polymorphisms and chemerin levels, whereas high chemerin levels were associated with obesity, female sex, diabetes mellitus, hypertension, current smoking, high platelet and leukocyte counts, anemia, impaired renal function, high C-reactive protein (CRP) levels, and multi-vessel disease. Kaplan–Meier survival curves indicated that the patients with high chemerin and CRP levels, but not those with *RARRES2* polymorphisms, had a lower survival rate and higher combined cerebral and cardiovascular event rates. Combined chemerin and CRP levels further revealed a stepwise increase in poor clinical outcomes from low- to high-risk subgroups. In conclusion, rs3735167 is the lead *RARRES2* polymorphism for chemerin levels in Taiwanese. Chemerin levels, but not the rs3735167 genotypes, predicted the long-term outcome of CAD, especially when combined with CRP levels.

## 1. Introduction

Chemerin, a novel adipokine highly expressed in the white adipose tissue, is associated with inflammation and adipogenesis, and also known as retinoic acid receptor responder protein 2 (*RARRES2*) [1,2,3,4]. Chemerin not only regulates the expression of adipocyte genes linked with glucose and lipid homeostasis but also affects innate and adaptive immunity as well as cascades of fibrinolytic, coagulation, and other inflammatory [3,5,6]. Plasma chemerin is increased in chronic inflammatory diseases, and elevated circulating chemerin levels is positively associated with detrimental effects in lipid, glucose and cytokine homeostasis, serving as a connection among obesity, metabolic disorders, and inflammation [7,8,9,10,11]. Furthermore, by promoting the formation of vascular inflammation through recruiting macrophages to inflamed blood vessels, chemerin may develop atherogenesis [12]. 

Using a genome-wide meta-analysis, Tönjes et al. [13] highlighted the aspect of *RARRES2* genetic variants in the control of circulating chemerin. Two other genome-wide association studies (GWASs) have indicated no genome-wide significant association between *RARRES2* genotypes and chemerin levels [14,15]. By Genotype-Tissue Expression (GTEx) data set, *RARRES2* SNPs were found associated with the expression quantitative trait loci of *RARRES2* and nearby genes, supporting the crucial roles of *RARRES2* genotypes [16]. Our preliminary analysis revealed that promoter polymorphisms of *RARRES2* were more significantly associated with circulating chemerin levels in a Taiwanese population [7]. The current study aimed to investigate the genetic basis of chemerin levels by conducting a GWAS in a Taiwan Biobank (TWB) population [17] and to confirm the crucial role of circulating chemerin levels and *RARRES2* polymorphisms in the long-term outcome of patients with angiographically confirmed coronary artery disease (CAD), especially when combined with C-reactive protein (CRP) level.

## 2. Results

### 2.1. Clinical and Biochemical Characteristics of TWB Participants and CAD Patients

Table 1 provides a summary of the baseline characteristics of the TWB participants and CAD population stratified by survival status in the follow-up period. Compared with the surviving CAD patients, those who died were older and have higher incidences of diabetes mellitus (DM), initial presentation other than stable angina pectoris, and multiple vessel disease; higher serum creatinine, CRP, and chemerin levels; higher leukocyte counts; and lower hematocrit and estimated glomerular filtration rates (eGFR). 

### 2.2. Results of GWAS and Replication Genotyping

In the present GWAS, we fitted a linear regression model for genotype trend effects. The peak of the –log_10_
*p* value for circulating chemerin was found on chromosome 7q36.1 where *RARRES2* is located. Eight SNPs passed the genome-wide significance threshold with each minor allele positively associated with circulating chemerin and rs3735167 was the most significant SNP (*p* = 2.35 × 10^−21^) (Figure 1A, Appendix A). Conditional analysis with further adjustment of the rs3735167 genotypes showed none of the SNPs around the *RARRES2* locus had significance *p* < 0.01 (Figure 1B, Appendix A), indicating that, in this chromosomal region, variances in chemerin concentrations were mainly explained by rs3735167. For replication, we further genotyped rs1962004 using the TaqMan assay in a previously reported cardiovascular health examination population [10] and by stepwise regression analysis, rs3735167 remained the only independent SNP associated with chemerin levels in this population (Appendix A). 

### 2.3. Associations Between Chemerin and CRP Levels and Clinical and Biochemical Correlations in the CAD Patients

After Bonferroni correction for multiple testing, significant correlations were observed between chemerin levels and BMI; hematocrit, leukocyte, and platelet counts; eGFR; and creatinine and CRP levels (Table 2). A positive association between chemerin and BMI was demonstrated in TWB participants (*p* = 1.0 × 10^−72^) and CAD population (*p* = 0.0004) respectively. Furthermore, associations between BMI and tertiles of circulating chemerin levels also showed consistent correlations in TWB participants (*p* = 1.17 × 10^−63^) and CAD population (*p* = 0.002). Significant correlations were also observed between CRP levels and hematocrit, leukocyte counts, and serum creatinine and chemerin levels. By analyzing the associations with risk factors for cardiovascular disease, plasma levels of chemerin were significantly higher in women, current smokers, those with hypertension, and those with DM (Appendix A). Plasma CRP levels were significantly higher in current smokers and those with DM.

### 2.4. Circulating Chemerin Levels, RARRES2 Genotypes, and Long-Term Prognosis in Patients with CAD

In the CAD population, the follow-up time was 1022 ± 320 days (minimal: 5 days; maximum: 1460 days) with 27 patients died during the follow-up. Using ROC curve analysis and the Youden index, the best prognostic cutoff values were 163.8 ng/mL and 9.7 mg/L, respectively, for chemerin and CRP levels. Kaplan–Meier survival analysis showed that a high chemerin level was a strong predictor of mortality (Figure 2A, *p* = 7.61 × 10^−7^) and a secondary endpoint (Figure 2B, *p* = 2.26 × 10^−9^), as well as a high CRP level was a strong predictor of mortality and a secondary endpoint (Figure 2C,D). When the CAD patients were divided into three subgroups according to chemerin and CRP levels, the combination of high chemerin and CRP levels demonstrated by Kaplan–Meier survival curves was a powerful predictor of all-cause death and secondary endpoints (*p* = 4.74 × 10^−16^ and *p* = 4.64 × 10^−13^, respectively; Figure 2E,F). Cox regression analysis indicated that higher circulating chemerin and CRP levels were the independent predictors of both primary and secondary endpoints (Table 3). When combined circulating chemerin and CRP levels were analyzed, a stepwise increase in poor clinical outcomes from low- to high-risk subgroups was noted. As shown in Appendix A, stepwise and significant increases in age, leukocyte and platelet counts, serum creatinine level, and frequency of DM, as well as stepwise decreases in eGFR and hematocrit, were demonstrated for each additional risk of subgroups. We further genotyped the three polymorphisms of rs3735167, rs1962004, and rs7806429 in the CAD population and found borderline significance between *RARRES2* polymorphisms and chemerin levels (minimal *p* = 0.038 for rs3735167; Table 2) and no significant difference between *RARRES2* genotypes and the long-term outcome of CAD patients (Appendix A).

## 3. Discussion

In this investigation, we confirmed that common variations near or within *RARRES2* were associated with plasma chemerin concentrations at a genome-wide significance with SNP rs3735167 to be the lead *RARRES2* polymorphism in a Taiwanese population. Furthermore, high chemerin and CRP levels and their combination are associated with the severity and a poor prognosis of CAD. This is the first report, to the best of our knowledge, to reveal that the synergistic effect of chemerin and CRP levels predict the long-term outcome of patients with angiographically confirmed CAD. By contrast, markedly decreased explained variance in chemerin levels in patients with CAD indicated that the effect of *RARRES2* polymorphisms was not large enough to alter the risk of mortality and secondary outcomes.

### 3.1. Chemerin Levels and the Long-Term Outcome of Various Disease States Including CAD

Chemerin has been suggested as a marker to predict cardiovascular risk [18] and several studies have shown that circulating chemerin concentrations correlated with various cardio-metabolic parameters and with CAD and the severity of atherosclerosis [18,19,20,21]. Gasbarino et al. [22] showed circulating chemerin is associated with carotid plaque instability. Leiherer et al. [15] found elevated plasma chemerin is correlated with renal impairment and is predictive for occurrence of cardiovascular episodes in patients that underwent angiography where half of their study patients had significant CAD. By contrast, all the study participants for the outcome study in our population were angiographically confirmed CAD patients. Our data demonstrated that a high chemerin level is associated with increased mortality and major adverse cerebral and cardiovascular events as well as multiple prognostic predictors of adverse outcomes. 

### 3.2. The Role of Chemerin in the Pathophysiology of CAD

Our data revealed that higher chemerin levels were found in CAD patients than in TWB population. Additionally, a recent prospective cohort study demonstrated a strong positive association with a clear dose-response trend between chemerin and myocardial infarction (MI), independent of established risk factors [23]. Participants who developed MI during follow-up had higher concentrations of chemerin than at study baseline. Immune-inflammatory responses have been increasingly proposed in the pathogenesis of atherosclerosis, which is found to be the leading cause of CAD [24]. Chemerin has been proposed to play a vital role in the pathophysiology of CAD by acting as a chemokine and an adipokine, involving mechanisms in more than one level of metabolic and immune-inflammatory processes [5]. It participates in activation and migration of immune cells to sites of injury on endothelium and smooth muscle cells [20,25]. Receptors of chemerin are identified on the endothelium of blood vessels and on their underlying smooth muscle layers [25]. The damage endothelium may uncover chemerin receptors on smooth muscle cells and cause atherosclerosis [20]. Chemerin activates the adhesion of macrophage to fibronectin and VCAM-1, and stimulates adhesion [12]. Secretion of chemerin by perivascular adipose tissue can result in contraction of vascular smooth muscle cells and acts as a link between chemerin and the development of hypertension [25]. Chemerin induces production of the adhesion molecules of ICAM1 and E-selectin and interacts with endothelium [26] to promote the releases of MMP which may play a role on blood vessel remodeling and growing in vitro experiments [14,27]. With the ability to regulate MMPs and other growth factors [27,28], chemerin could involve in the progression and the development of thrombus or embolus. Furthermore, chemerin activates apoptosis in a time- and dose-dependent way in cultured cardiomyocytes, which plays a vital role in the pathophysiological development of diverse heart diseases including CAD, acute myocardial infarction and congestive heart failure [29,30,31]. By acting as an adipokine, chemerin has an established detrimental role in metabolic disorders [32]. Chemerin affects the lipid [3] and glucose metabolism [33] possibly by changing their infiltration into endothelium, these are additional properties of chemerin linked to the pathogenesis of CAD. These observations suggested that chemerin related metabolic and immune-inflammatory pathways are crucial in the pathogenesis of CAD.

### 3.3. Combining Biomarkers and Risk Scores for the Prognosis of CAD

Multiple marker approaches with or without biomarker scores have improved risk estimations for cardiovascular events in healthy cohorts and patients with acute coronary syndrome [34,35,36]. By evaluating multiple biomarkers of cardiovascular stress, Sabatine et al. [35] found that the approach helped to select those patients with stable coronary disease who were at a higher possibility of heart failure and cardiovascular death, which may be beneficial for identifying patients who obtain compelling advantages from angiotensin-converting enzyme inhibitor treatment. Wang [37] suggested that finding “uncorrelated” biomarkers outside of an already characterized pathway may improve the performance of risk models. Our data showed that circulating chemerin and CRP levels are pathobiologically diverse biomarkers with fair correlations in CAD patients. A combination of these two biomarkers has been found to be associated with multiple risk pathways with synergistic effects in predicting the long-term outcome of angiographically confirmed CAD.

### 3.4. Lead SNP of RARRES2 Polymorphisms for Chemerin Levels

Previous GWASs derived from diverse Caucasian populations have shown variable results on the association between *RARRES2* polymorphisms and circulating chemerin levels [13,14,15]. Tönjes et al. [13] provided the only report in Caucasians revealing genome-wide significant association between *RARRES2* locus and chemerin levels with the rs7806429 in the 3′ untranslated region as the lead SNP. This is in contrast with our GWAS from the TWB population, in which the rs3735167 polymorphism, located −781 base pair upstream of the transcriptional initiation site of *RARRES2*, is the lead SNP for chemerin levels. These differences may attribute to ethnic genetic heterogeneity in the association of *RARRES2* SNPs with chemerin levels; each ethnic group may present specific results. The associations were further confirmed in two other Taiwanese populations, one from a cardiovascular health examination and another from CAD patients. In this study, we also found a markedly diminished effect of *RARRES2* SNPs on chemerin levels in CAD patients when compared with the healthy populations (Appendix A). This may at least partly explain why controversial results were noted in previous GWASs. The diminished effect of *RARRES2* SNPs may also explain why circulating chemerin levels, but not the lead *RARRES2* polymorphism, predict the long-term outcome of angiographically confirmed CAD. The results suggested that the GWAS result from a healthy population may not be directly applied to the disease population such as CAD.

### 3.5. Limitations of the Study

This study has several limitations. First, only a medium-sized CAD population was analyzed with a follow-up of a moderate duration and low mortality. A larger population with a longer follow-up may further confirm the associations and roles of multiple markers, thereby facilitating predicting the risk of angiographically confirmed CAD. Second, more than 80% of the patients presented with stable angina pectoris, and only 12% presented with acute coronary syndrome or congestive heart failure. Thus, patients with chronic stable ischemic heart disease constituted most of the study population. Although significantly higher mortality was noted in patients with acute cardiac disease, the adjustment of the clinical presentation did not attenuate the significance of chemerin and CRP levels and their combination in the prognosis, suggesting the crucial role of both biomarkers in the long-term outcome of patients with CAD.

## 4. Materials and Methods

### 4.1. Participants

The GWAS cohort consisted of participants from the TWB population. Information was gathered at recruitment centers across Taiwan between 2008 and 2015. A total of 2349 participants with no history of cancer, stroke, CAD, or systemic disease were recruited. Exclusion criteria were subjects who announced to withdraw the informed consent (*n* = 2), fasting for <6 h (*n* = 38), no chemerin level available (*n* = 1), no rs3735167 data available (*n* = 1), and quality control (QC) for GWAS (*n* = 110); finally, 2197 participants were enrolled for the analysis. Ethical approval (approval number: 05-X04-007) was received from the Research Ethics Committee of Taipei Tzu Chi Hospital, Buddhist Tzu Chi Medical Foundation, and Ethics and Governance Council of the Taiwan Biobank (approval number: TWBR10507-02 and TWBR10611-03). Each participant signed an approved informed consent form.

Between July 2010 and September 2013, a total of 565 patients with CAD who presented with more than ≥50% stenosis of one major coronary artery and performed coronary angiography were enrolled from National Taiwan University Hospital. A flow chart of the study inclusion and exclusion criteria and the definition of baseline measurements were previously reported [38] and finally 481 patients were enrolled. From the patients’ medical records, all clinical data were collected. All-cause mortality was the primary endpoint. The secondary endpoint was defined as the combination of all-cause death, myocardial infarction, stroke, and hospitalization for heart failure. After a follow-up period of 1022 ± 320 days, 27 patients died. Seven patients who were lost during follow-up after recruitment were called by telephone before the completion of the research. Three of them had expired; the causes of death were confirmed by the family members. Approval was collected from the Institutional Review Board of National Taiwan University Hospital (No.201002015M). Written informed consent was provided to all of the participants.

### 4.2. Genomic DNA Extraction and Genotyping

For the TWB participants, DNA was isolated from blood samples using a PerkinElmer chemagic™ 360 instrument following the manufacturer’s instructions (PerkinElmer, Waltham, MA, USA). SNP genotyping was conducted using custom TWB chips and performed on the Axiom Genome-Wide Array Plate System (Affymetrix, Santa Clara, CA, USA) [17]. For the CAD population, genotyping was completed adopting TaqMan SNP Genotyping Assays of Applied Biosystems (ABI; Foster City, CA, USA) [38,39].

### 4.3. GWAS Analysis

For GWAS analysis, each genomic DNA was genotyped using the Axiom TM-TWB genome-wide array comprising 642,832 single-nucleotide polymorphisms (SNPs) with minor allele frequencies of ≥5% in a set of 1950 samples from a Taiwanese Han Chinese population [17]. Further, SNP rs3735167, previously reported to be the most significant SNP associated with chemerin levels [7], was also genotyped with the Taqman Assay. In this investigation, all the samples enrolled for the analysis had a call rate of ≥97%. SNP QC was set as follows: An SNP call rate of <3%, a minor allele frequency of <0.05, and a violation of Hardy–Weinberg equilibrium (*p* < 10^−6^); these were excluded from subsequent analyses. After QC, a total of 614,820 SNPs were enrolled for the GWAS analysis.

### 4.4. Laboratory Examinations

By adopting ELISA kits (R&D, Minneapolis, MN, USA), circulating plasma levels of chemerin were determined. Circulating plasma levels of CRP were measured using the particle-enhanced turbidimetric immunoassay technique (Siemens Healthcare Diagnostics Ltd., Camberley, UK). The increase in turbidity that accompanies aggregation is proportional to the CRP concentration.

### 4.5. Statistical Analysis

Continuous variables were examined utilizing analysis of variance or a two-sample t-test, and are presented as the mean ± standard deviation, whereas median and interquartile ranges are given when the distribution was strongly skewed. Differences in categorical data distribution were identified by adopting chi-squared test or chi-squared test for trend. To conform to a normality assumption, serum creatinine and fasting plasma glucose levels and fasting plasma CRP and chemerin levels were logarithmically transformed before investigation. A generalized linear model was adopted to examine the relationship of chemerin with the analyzed genotypes and confounders. The genetic effect was assumed to be additive, and adjustments were made for sex, age, body mass index (BMI), and present status of smoking. Genome-wide scans were calculated using the analysis software package PLINK. *p* values below the threshold of *p* = 5 × 10^−8^ were considered genome-wide significant. Conditional analysis in GWAS was conducted by adding the most strongly associated SNP into the regression model as a covariate and by testing the residual association with all remaining SNPs.

We compared CRP and chemerin levels and the rs3735167 genotypes to predict primary and secondary endpoints by plotting curves of receiver operating characteristic (ROC). Subsequently, the area under the ROC curve (AUC) for all variables of interest was compared non-parametrically. A survival curve was identified adopting the Kaplan–Meier estimate, and significance was examined adopting the log-rank method. All calculations were performed using SPSS version 22 (SPSS, Chicago, IL, USA).

## 5. Conclusions

Our data revealed rs3735167 to be the lead *RARRES2* polymorphism for chemerin levels in a Taiwanese population. Chemerin levels, but not the rs3735167 genotype, predict the long-term outcome of patients with angiographically confirmed CAD, especially when combined with CRP levels.

## Figures and Tables

**Figure 1 ijms-20-01174-f001:**
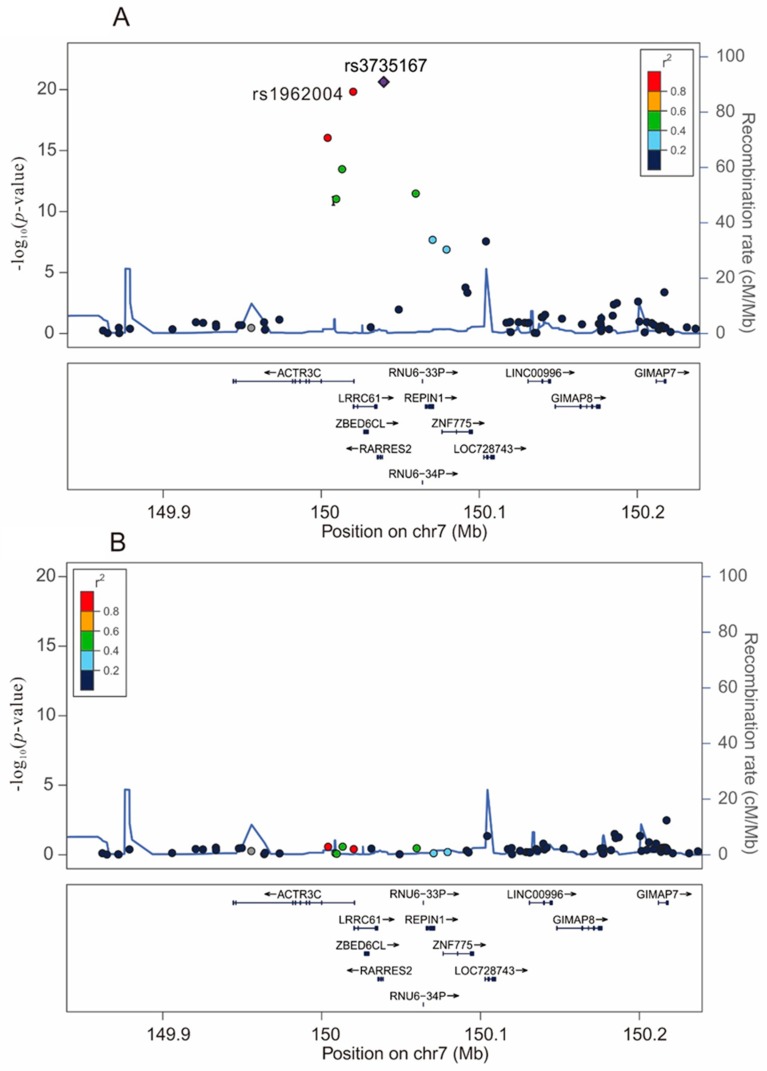
Regional association plots at a region of 100 kb surrounding the *RARRES2* locus on chromosome 7. Regional association plots for the top-hit of association with chemerin levels at a region of 100 kb surrounding the *RARRES2* locus on chromosome 7, without (**A**) or with (**B**) conditional analysis with adjustment of the rs3735167 polymorphism.

**Figure 2 ijms-20-01174-f002:**
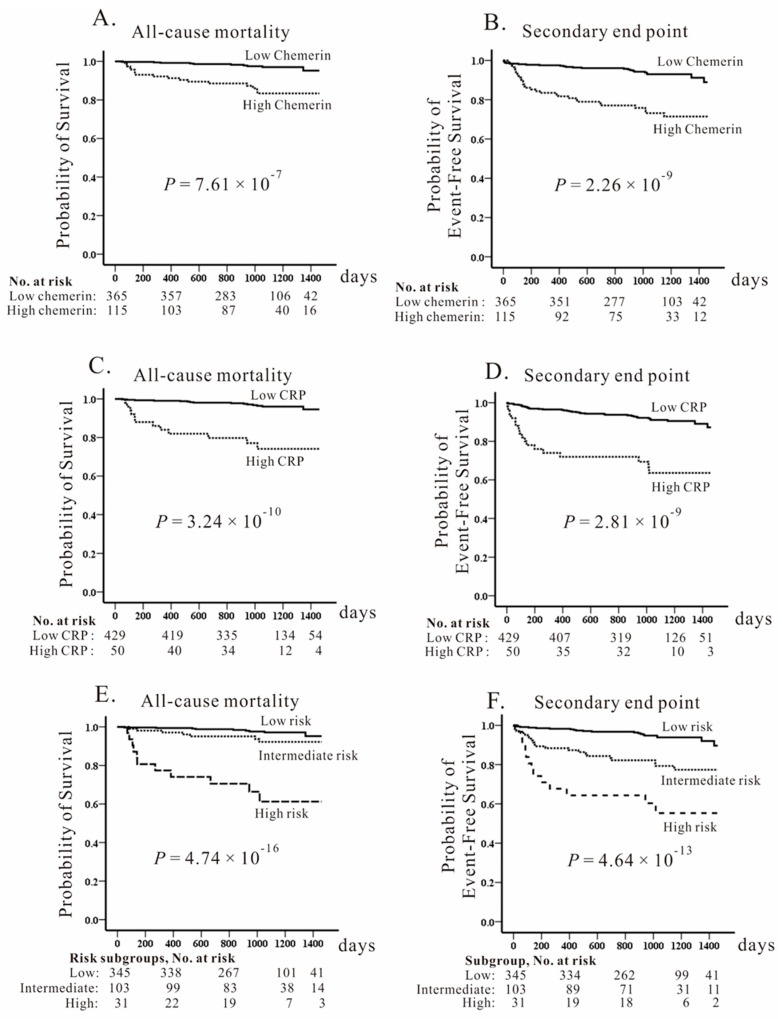
Kaplan–Meier curves of the cumulative incidence of primary and secondary endpoints. Individuals are stratified according to chemerin levels (>163.8 ng/mL vs. ≤163.8 ng/mL) (**A**,**B**) and C-reactive protein (CRP) levels (>9.7 mg/L vs. ≤9.7 mg/L) (**C**,**D**) as well as their combination (**E**,**F**) in patients with angiographically confirmed coronary artery disease (CAD). Significantly higher mortality and combined endpoints for CAD were noted for higher chemerin and CRP levels as well as higher risk subgroups of combined chemerin/CRP levels. The study patients were followed for 1022 ± 320 days.

**Table 1 ijms-20-01174-t001:** Clinical and biochemical characteristics of the Taiwan Biobank (TWB) participants and coronary artery disease (CAD) patients according to their survival state.

	TWB (2197)	CAD		
		Survival (454)	Mortality (27)	*p* value ^a^
Baseline characteristics				
Sex (male/female)	984/1213	370/84	18/9	0.65
Age (years)	48.4 ± 10.9	64.9 ± 11.0	77.1 ± 9.3	<0.0001
Body mass index (kg/m2)	24.2 ± 3.5	26.0 ± 4.0	25.2 ± 4.2	0.56
Hypertension (%)	15.6	77.8	85.2	0.58
Diabetes mellitus (%)	5.9	43.2	63.0	0.02
Dyslipidemia (%)	48.5	61.7	48.1	0.90
Current smoker (%)	18.0	24.7	18.5	0.80
Initial presentation				
Stable angina pectoris (%)		87.4	29.6	<0.0001
ACS/MI (%)		5.7	40.7	
CHF/lung edema (%)		3.5	22.2	
Others (%)		3.3	7.4	
CAD (S vs. D vs. T) (%)		29.3:28.6:42.1	3.7:18.5:77.8	0.004
Biochemistry				
Serum creatinine (mg/dL)	0.7 (0.6–0.9)	1.1 (0.9–1.3)	1.4 (1.1–2.2)	0.007
eGFR	108.0 ± 25.0	71.0 ± 23.7	46.7 ± 26.0	0.0004
Blood cell counts				
Leukocyte counts (10^3^/μL)	6.1 ± 1.6	6.5 ± 1.8	8.3 ± 4.8	0.0007
Hematocrit (%)	43.9 ± 4.6	41.1 ± 5.1	35.4 ± 7.2	0.0008
Platelet counts (10^3^/μL)	240.1 ± 56.4	213.5 ± 60.0	185.4 ± 70.3	0.29
Inflammation markers				
C-reactive protein (mg/L)		2.4 (1.2–4.1)	4.2 (2.2–24.7)	0.0002
Chemerin (ng/mL)	96.6 (80.6–110.3)	123.3 (93.8–157.1)	176.2 (108.5–227.6)	0.001

ACS/MI: Acute coronary syndrome or myocardial infarction; CHF: Congestive heart failure; S vs. D vs. T: Single vs. double vs. triple vessel coronary artery disease; eGFR: estimated glomerular filtration rate; Data are expressed as mean ± SD, percentage, or median (interquartile range) as appropriate. A Comparison between CAD patients according to their survival state.

**Table 2 ijms-20-01174-t002:** Association between circulating chemerin and CRP levels and measurable cardiovascular risk factors in patients with coronary artery disease.

		Chemerin			CRP		
		r	*p* value ^a^	Adjusted *p* value ^b^	r	*p* value	Adjusted *p* value
Anthropology	Age (years)	0.011	0.803		0.091	0.046	
	Body mass index (kg/m^2^)	0.160	0.0004	0.003	0.054	0.242	
Blood cell counts	Leukocyte counts (10^3^/μL)	0.262	<0.0001	<0.0001	0.408	<0.0001	<0.0001
	Hematocrit (%)	−0.382	<0.0001	<0.0001	−0.173	0.0002	0.002
	Platelet counts (10^3^/μL)	0.200	<0.0001	0.0002	0.074	0.107	
Renal function	Serum creatinine (mg/dL)	0.470	<0.0001	<0.0001	0.148	0.001	0.009
	eGFR (mL/min/1.86 m^2^)	−0.553	<0.0001	<0.0001	−0.11	0.017	
Inflammatory marker	CRP (mg/L)	0.378	<0.0001	<0.0001			
	Chemerin (ng/mL)				0.378	<0.0001	<0.0001

Abbreviations as in Table 1. ^a^
*p* value: Adjusted for sex and age. ^b^ Adjusted *p* value: After Bonferroni correction; a Bonferroni correction for multiple testing was used with α = 0.005 after the nine different tested laboratory variables were considered. Only significant *p* values of <0.05 are shown.

**Table 3 ijms-20-01174-t003:** Predictors of primary and secondary endpoints in Cox regression analysis.

	Predictors		Model 1 ^a^	Model ^b^	Model ^c^
Primary end point	Chemerin level subgroups ^d^	Hazard ratio (95% CI)	5.71 (2.62–12.48)	4.55 (1.86–11.16)	3.55 (1.46–8.68)
		*p* value	<0.0001	0.001	0.005
	CRP level subgroups ^e^	Hazard ratio (95% CI)	7.82 (3.66–16.71)	5.73 (2.39–13.75)	4.27 (1.72–10.61)
		*p* value	<0.0001	<0.0001	0.002
	Combined risk subgroups (intermediate vs. low)	Hazard ratio (95% CI)	2.61 (0.97–7.00)	2.72 (0.94–7.93)	1.85 (0.62–5.53)
		*p* value	0.057	0.063	0.275
	Combined risk subgroups	Hazard ratio (95% CI)	17.02 (7.04–41.13)	11.17 (3.84–32.47)	8.71 (2.99–25.31)
	(high vs. low)	*p* value	<0.0001	<0.0001	<0.0001
Secondary end point	Chemerin level subgroups	Hazard ratio (95% CI)	4.44 (2.59–7.60)	3.78 (2.11–6.76)	3.04 (1.69–5.47)
		*p* value	<0.0001	<0.0001	0.0002
	CRP level subgroups	Hazard ratio (95% CI)	4.84 (2.72–8.60)	3.78 (2.02–7.07)	2.76 (1.45–5.25)
		*p* value	<0.0001	<0.0001	0.002
	Combined risk subgroups	Hazard ratio (95% CI)	3.82 (2.07–7.05)	4.14 (2.17–7.89)	3.18 (1.64–6.18)
	(intermediate vs. low)	*p* value	<0.0001	<0.0001	0.001
	Combined risk subgroups	Hazard ratio (95% CI)	9.47 (4.70–19.06)	5.87 (2.67–12.93)	4.52 (2.04–10.03)
	(high vs. low)	*p* value	<0.0001	<0.0001	0.0002

95% CI: 95% confidence interval ^a^ Model 1: Unadjusted. ^b^ Model 2: Adjusted for baseline data (sex, age, BMI, current smoking status, diabetes mellitus, hypertension, and dyslipidemia). ^c^ Model 3: Adjusted for baseline data and initial presentation (sex, age, BMI, current smoking status, diabetes mellitus, hypertension, dyslipidemia, and initial presentation). ^d^ Chemerin level subgroups: >163.8 ng/mL vs. ≤163.8 ng/mL of chemerin level. ^e^ CRP level subgroups: >9.7 mg/L vs. ≤9.7 mg/L of CRP level.

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
