# Peer review of "Circulating Chemerin Levels, but not the RARRES2 Polymorphisms, Predict the Long-Term Outcome of Angiographically Confirmed Coronary Artery Disease"

_ijms, 2019, doi:10.3390/ijms20051174_

Reviewer 1 Report

The limit of this study beyond that described by the authors themselves is the need to confirm these results with biochemical assays as RT-PCR and Immunoblotting analysis. Therefore, CRP or Chemerin levels should also be measured with other techniques, RT-PCR or Immunoblotting. Furthermore, there are too many results in supplementary materials, that are moreover not visible.

Author Response

Reviewer 1
1. The limit of this study beyond that described by the authors themselves is the need to confirm these results with biochemical assays as RT-PCR and Immunoblotting analysis. Therefore, CRP or chemerin levels should also be measured with other techniques, RT-PCR or Immunoblotting.
Answer:
We are very grateful for the reviewer’s valuable comments. Measurement of tissue expression is impractical in the clinical setting at present and circulating biomarker level is the most valuable tool in daily practice. However, we agree that our investigation can be extended by biochemical assays such as RT-PCR and immunoblotting analysis in the cellular or tissue level. We would like to explore the role of chemerin in the physiopathology of cardiac diseases by such approach in the future study.  

2. Furthermore, there are too many results in supplementary materials, that are moreover not visible.
Answer:
We are very grateful for the reviewer’s valuable comments. We have contacted editor and she promised that the supplementary materials will be sent to reviewers.

Reviewer 2 Report

The paper by Er et al. investigates circulating chemerin levels as a predictor for clinical outcomes in confirmed coronary artery disease. Papel is well-written and results clearly presented. There are some minor issues that need to be clarified.

The short follow-up time is a clear limitation of the study. Although authors acknowledged this in the discussion, the actual follow-up time has to be clearly stated in results, methods, and figures.

The findings suggesting that chemerin levels are a good prognostic indicator for mortality in coronary artery disease are well supported; however, it is important to acknowledge that this is base on a single measurement. Was this at the start of the study? What are the kinetics of chemerin levels during CAD pathogenesis? Has somebody looked into this? If not, can authors speculate if this is important?

Is chemerin a simple biomarker or does it have any role during CAD pathogenesis? Authors mentioned that chemerin activates apoptosis of cardiomyocytes in discussion, which is relevant. Please expand about the biological role of circulating chemerin in the context of CAD.

The GWAS data should be submitted to a public database repository such as GWAS Central or GWAS Catalog, and accession number provided in manuscript.

It might be interesting to show in supplementary data the correlation graph for chemerin and BMI, considering that chemerin is produced by white adipose tissue.

Author Response

Reviewer 2
1. The short follow-up time is a clear limitation of the study. Although authors acknowledged this in the discussion, the actual follow-up time has to be clearly stated in results, methods, and figures.
Answer: 
We would like to thank the reviewer for valuable comments. The actual follow-up time was 1022 ± 320 days (minimal: 5 days; maximum: 1460 days) with 27 patients died during the follow-up. The follow-up time has been clearly stated in results, methods, and figures.

2. The findings suggesting that chemerin levels are a good prognostic indicator for mortality in coronary artery disease are well supported; however, it is important to acknowledge that this is base on a single measurement. Was this at the start of the study? What are the kinetics of chemerin levels during CAD pathogenesis? Has somebody looked into this? If not, can authors speculate if this is important?
Answer:
Thank you for your suggestion. We measured chemerin levels only at the start of the study just before the cardiac catheterization for CAD patients. The average circulating chemerin levels was 96.6ng/mL in TWB participants and 136.3ng/mL in CAD patients respectively. Significant difference was observed between TWB population and CAD patients (P=3.2 × 10-28) even after adjusting for age, sex and BMI. We suggested that relatively healthy adults such as population of TWB presented lower circulating chemerin levels than in CAD patients. The kinetic changes of circulating chemerin levels with time and the progression of the CAD are uncertain in the literature. A recent prospective cohort study demonstrated a strong positive association with a clear dose-response trend between chemerin and myocardial infarction (MI), independent of established risk factors [23]. Participants who developed MI during follow-up had higher concentrations of chemerin than at study baseline. These observations suggested that chemerin related immune-inflammatory pathways may play a role in the pathogenesis of CAD

3. Is chemerin a simple biomarker or does it have any role during CAD pathogenesis? Authors mentioned that chemerin activates apoptosis of cardiomyocytes in discussion, which is relevant. Please expand about the biological role of circulating chemerin in the context of CAD.
Answer:

Thanks for your comments. We have expanded the discussion related to the biological role of circulating chemerin in the context of CAD at “Discussion” section and under the paragraph of 3.2.” The role of chemerin in the pathophysiology of CAD”
Immune-inflammatory responses have been increasingly proposed in the pathogenesis of atherosclerosis, which is found to be the leading cause of CAD [24]. Chemerin plays a vital role in the pathophysiology of CAD by acting as a chemokine and an adipokine, involving mechanisms in more than one level of metabolic and immune-inflammatory processes [5]. It participates in activation and migration of immune cells to sites of injury on endothelium and smooth muscle cells [20, 25]. Receptors of chemerin are identified on the endothelium of blood vessels and on their underlying smooth muscle layers [25]. The damage endothelium may uncover chemerin receptors on smooth muscle cells and causing atherosclerosis [20]. Chemerin activates the adhesion of macrophage to fibronectin and VCAM-1, and stimulated adhesion [12]. Secretion of chemerin by perivascular adipose tissue can result in contraction of vascular smooth muscle cells and acts as a link between chemerin and the development of hypertension [25]. Chemerin induces production of the adhesion molecules of ICAM1 and E-selectin and interacts with endothelium [26] to promote the releases of MMP which may play a role on blood vessel remodeling and growing in vitro experiments [14, 27]. With the ability to regulate MMPs and other growth factors [27, 28], chemerin could involve in the progression and the development of thrombus or embolus. Furthermore, chemerin activates apoptosis in a time- and dose-dependent way in cultured cardiomyocytes, which plays a vital role in the pathophysiological development of diverse heart diseases including CAD, acute myocardial infarction and congestive heart failure [29-31]. By acting as an adipokine, chemerin has an established detrimental role in metabolic disorders [32]. Chemerin affects the lipid [3] and glucose metabolism [33] possibly by changing their infiltration into endothelium, these are additional properties of chemerin linked to the pathogenesis of CAD.

4. The GWAS data should be submitted to a public database repository such as GWAS Central or GWAS Catalog, and accession number provided in manuscript.
Answer:
We would like to thank the reviewer for the valuable suggestion. GWAS Central or GWAS Catalog reveal that they can only include data in the Catalog or database only if the data has been published in a peer-reviewed journal and indexed in PubMed. They aren’t currently able to provide accession number before publication. However, they also mention that we can contact them if we have an accepted publication for which we need to include a data availability statement in our manuscript, we then can submit summary statistics to the GWAS Catalog or GWAS Central. Therefore, we are very pleased to submit and register our GWAS data on GWAS Central or GWAS Catalog when our manuscript became accepted. 

5. It might be interesting to show in supplementary data the correlation graph for chemerin and BMI, considering that chemerin is produced by white adipose tissue.
Answer:
Thank you for your kindly comment. We have presented the correlation graph for chemerin and BMI in supplementary materials (Supplementary Figure 2). Our study demonstrated a strong positive association between chemerin and BMI in TWB participants (P=1.0 × 10-72) and CAD population (P=0.0004) which is in accordance to other prior studies [32]. Furthermore, associations between BMI and tertiles of circulating chemerin levels also showed consistent results in TWB participants (P=1.17 × 10-63) and CAD population (P=0.002).

Association between BMI and circulating chemerin levels in TWB population and patients with angiographically confirmed coronary artery disease respectively (A, B); Association between BMI and tertiles of circulating chemerin levels were presented as Box and whisker plots analysis (C, D). P value: adjusted for sex and age. Abbreviations: T, tertiles and as in Table 1

Round  2

Reviewer 1 Report

The new version of the manuscript with supplementary data is clearer. It is a preliminary study but it can give a new starting point to the scientific community.